# Mandibular Endochondral Growth Is Specifically Augmented by Nutritional Supplementation with Myo-Inositol Even in Rabbits

**DOI:** 10.3390/dj12030049

**Published:** 2024-02-26

**Authors:** Miho Shimoyama, Hiroyuki Kanzaki, Syunnosuke Tohyama, Tomomi Ida, Misao Ishikawa, Yuta Katsumata, Chihiro Arai, Satoshi Wada, Shugo Manase, Hiroshi Tomonari

**Affiliations:** 1Department of Orthodontics, School of Dental Medicine, Tsurumi University, Yokohama 230-8501, Japan; smymmh@outlook.jp (M.S.); toyama-s@tsurumi-u.ac.jp (S.T.); idatomomi.jelico@gmail.com (T.I.); yutakatsumata0904@gmail.com (Y.K.); arai-chihiro@tsurumi-u.ac.jp (C.A.); s063082@yahoo.co.jp (S.M.); tomonari-h@tsurumi-u.ac.jp (H.T.); 2Department of Anatomy, School of Dental Medicine, Tsurumi University, Yokohama 230-8501, Japan; ishikawa-misao@tsurumi-u.ac.jp; 3Department of Oral and Maxillofacial Surgery, Kanazawa Medical University, Kanazawa 920-0293, Japan; wada-s@kanazawa-med.ac.jp

**Keywords:** mandibular retrognathism, endochondral growth, mandibular condylar cartilage, myo-inositol, pik3cd, functional appliances, twin block, bionator, activator

## Abstract

Mandibular retrognathism occurs by insufficient mandibular growth and causes several issues, such as respiratory difficulty and diminished masticatory function. At present, functional orthodontic appliances are used for stimulating mandibular growth in pediatric cases. However, the effectiveness of functional appliances is not always stable in daily practices. A more effective, reliable, and safer therapeutic method for mandibular growth promotion would be helpful for growing mandibular retrognathism patients. As we previously discovered that nutritional supplementation of myo-inositol in growing mice specifically increases mandibular endochondral growth, we performed preclinical animal experiments in rabbits in this study. Briefly, six-week-old male Japanese white rabbits were fed with or without myo-inositol supplementation in laboratory chow until 25 weeks old, and 3D image analysis using micro CT data and histological examinations was done. Myo-inositol had no systemic effect, such as femur length, though myo-inositol specifically augmented the mandibular growth. Myo-inositol increased the thickness of mandibular condylar cartilage. We discovered that the nutritional supplementation of myo-inositol during the growth period specifically augmented mandibular growth without any systemic influence, even in rabbits. Our results suggest the possibility of clinical use of myo-inositol for augmentation of the mandibular growth in growing mandibular retrognathism patients in the future.

## 1. Introduction

Mandibular retrognathism arises from insufficient mandibular growth, culminating in diminutive and retruded mandibular morphology [1,2]. It can lead to several issues, such as respiratory difficulty, temporomandibular joint disorders, diminished masticatory function, and esthetic concerns [3,4,5]. Orthodontic camouflage treatment or orthognathic surgical treatment is chosen in adult cases [6]. In some cases, non-surgical treatments such as mandibular advancement oral appliances and nasal positive air pressure (nCPAP) are used to improve obstructive sleep apnea (OSA) symptoms [7]. The American Association of Orthodontists released guidance to practicing orthodontists on the suggested role of the specialty of orthodontics in the management of obstructive sleep apnea [8], in which it announced that OSA can have many serious consequences if left untreated, and orthodontists should consider incorporating OSA screening for their patients. On the other hand, the growth promotion of the mandible is achieved by the use of functional orthodontic appliances in pediatric cases [9].

A functional appliance is reported to induce mandibular growth by augmentation of endochondral growth in mandibular condylar cartilage and remodeling of the temporomandibular joint [10,11]. In addition to skeletal change, dentoalveolar change also contributes to improving the retruded lower arch, Class II malocclusion [12]. In other words, functional appliances appear to be effective in improving Class II malocclusion in the short term by modifying both dentoalveolar and skeletal relationships. On the other hand, there are contradictory reports on the effects of functional appliances in the long term. Cacciatore et al. [13] reported that Class II malocclusion was corrected even in long-term evaluation judged by the value of Wits appraisal, which evaluates intermaxillary relationships based on the occlusal plane [14]. However, the long-term improvement of Class II malocclusion was relatively weak, judged by the Pogonion to Nasion perpendicular, which exhibits the relationship between the chin and cranial base [13]. A clinical trial revealed that the treatment timing for functional appliances influences long-term skeletal change [15]. Together, functional appliances give stable and effective results in the short term, though the improvement of Class II malocclusion in the long term was controversial in terms of skeletal effects.

In addition to the issue mentioned above, a functional appliance has another possible drawback because it is a removal appliance. The effect of functional appliances depends on the degree of cooperation and compliance [16,17]. Therefore, removable functional appliances could only obtain dentoalveolar correction of Class II malocclusion without clinically significant skeletal changes [18]. Judged by these circumstances, the efficacy of functional appliances may not consistently manifest in routine clinical applications [19], which demands a more efficient, dependable, and secure therapeutic approach to stimulate mandibular growth.

The primary determinant of mandibular growth predominantly relies on the process of endochondral ossification in mandibular condylar cartilage [20]. Endochondral ossification is modulated by various determinants, including genetic factors [21], hormonal regulation [22], and nutritional aspects [23]. Among these factors, growth factors such as insulin-like growth factor 1 (IGF-1) [24,25,26] or growth hormone [27] were used to induce mandibular growth by local injection into temporomandibular joint cavity in experimental animals. Because these factors augment cartilage growth not only in mandibular condylar cartilage but also in the growth cartilage of limbs, local injection into the temporomandibular joint cavity is required to obtain specific mandibular growth [28]. This topical application makes it difficult for clinical use.

As to nutritional factors which augment chondrocytic growth and differentiation, lipophilic vitamins [29], glucosamine and chondroitin sulfate [30], glucose and glucose-derived sugars [31], and vitamin D [32] were reported factors. Regarding the correlation between nutrition and mandibular growth, it was observed that murine subjects deficient in myo-inositol synthase and inositol monophosphatase exhibited pronounced mandibular retrognathism [33]. In the report, the administration of maternal mice with supplementary myo-inositol successfully ameliorated mandibular retrognathism among neonatal mice. Myo-inositol is one of the sugar alcohols with a small molecular weight. These results imply that the addition of myo-inositol could potentially enhance the process of endochondral growth in the mandible. 

We previously reported that nutritional supplementation of myo-inositol in growing mice specifically increases mandibular endochondral growth [34]. In the report, we further examined the mechanisms underlying the specific augmentation of mandibular endochondral growth and found that phosphatidylinositol 3-kinase catalytic delta polypeptide (PIK3CD) is the key enzyme to produce phosphatidylinositol from myo-inositol [35,36]. Furthermore, myo-inositol augments not only chondrocytic growth but also chondrocytic differentiation in cultured Pik3cd-expressing chondrocytes to a similar extent by BMP4 [37].

In this manuscript, we examined the extent of the augmentation in mandibular growth by myo-inositol supplementation in rabbits with future clinical applications in mind.

## 2. Materials and Methods

### 2.1. Animal Experiment with Rabbits

The protocols for animal experiments were reviewed and approved by the Institutional Animal Care and Use Committee of Tsurumi University (Nos. 20A009 and 29A053), and animal experiments were performed in compliance with the Regulation for Animal Experiments and Related Activities at Tsurumi University and the Animal Research: Reporting of In Vivo Experiments (ARRIVE) guidelines for preclinical studies. The animals used in our experiments were six-week-old male Japanese white (JW) rabbits (Tokyo Laboratory Animals Science Co., Ltd., Tokyo, Japan). We purchased ten JW rabbits at five weeks old, which first stayed in the facility for 1 week as an acclimation period. They were pooled and randomized, then divided into two groups (*n* = 5 each): control and myo-inositol group. As to the number of samples for experiments, we performed a power analysis using a calculated effect size (2.58) from previous data [34]; power analysis in the condition of effect size = 2.58, alpha error probability = 0.05, power = 0.8, which gave the required sample size as 4.

Rabbits in the control group were fed a normal diet (myo-inositol content: 9.5 mg/kg; Labo R Grower, Nosan Corp., Yokohama, Japan), and the myo-inositol group was fed a diet supplemented with myo-inositol (myo-inositol content: 66 g/kg; Fujifilm Wako chemicals, Osaka, Japan). Experimental animals were housed under a 12 h light/dark cycle with ad libitum access to food and deionized water during the experimental period. The facility maintains appropriate temperature, humidity, ventilation, and lighting. Also, it has a structure and strength to prevent the escape of experimental animals and prevent odors, noise, and waste from adversely affecting the surrounding environment.

To eliminate the possible influence of mastication of the laboratory chow on mandibular growth [38,39], the diet of both groups was mixed with water, which made the diet paste form and mastication unnecessary. All rabbits were fed the above diet from 6 to 25 weeks old. Body weight was measured once a week during the experimental period. 

Rabbits were anesthetized by CO_2_ inhalation and euthanized by intraperitoneal injection of over-dose pentobarbital (200 mg/kg; Nacalai Tesque, Inc., Kyoto, Japan) at 25 weeks old. The maxilla, the mandible, and the femur were excised, and the specimens were fixed with 4% paraformaldehyde in phosphate-buffered saline (PBS) overnight. These specimens were then scanned with Micro computed tomography (micro CT) and used for histological examinations.

### 2.2. Micro Computed Tomography Analysis

Specimens were scanned with a micro CT system (InspeXio SMX-225 CT; Shimadzu Corp., Kyoto, Japan) at the condition of the X-ray tube current = 70 mA, tube voltage = 160 kV, slice thickness = 0.611 mm, with remaining blinding by handling the data by unique numbering. Reconstituted DICOM data were then rendered into 3D images using RadiAnt Dicom viewer (Medixant Inc., Poznań, Poland) and analyzed with multi-planar reconstruction (MPR) mode. Three screen settings, XY-plane, YZ-plane, and ZX-plane, were used for the analysis. The length of the mandible, maxilla, and femur were measured in MPR mode. We used the following reference points for measurements. Cd: the condylar maximal projecting point of the mandibular angle. Al: the mesial alveolar bone apex of the mandibular first molar. Mu3: the distal alveolar bone apex of the maxillary third molar. Iu: the palatal alveolar bone apex of the maxillary incisor. Trochanteric fossa (Tf), and patellar surface of femur (Ps). The distance between Cd to Al, Mu3 to Iu, and Tf to Ps were measured as the length of the mandible, maxilla, and femur, respectively. The measurement errors for the mandibular length were 0.46 mm when a single investigator measured the same sample at a 2-week interval.

### 2.3. Preparation of Histological Sections

The rabbit’s specimen was decalcified with Morse’s solution (Fujifilm Wako chemicals) for six weeks. Then, specimens were washed with PBS, dehydrated with Et-OH, and embedded in paraffin. Serial histological sections (7 µm thick) were then prepared. Femur sections were prepared along with the longitudinal axis of the femur. The sections of the mandibular condyle were prepared along with the longitudinal axis of the condylar neck, including the sagittal plane.

### 2.4. Histological Analysis for the Thickness of the Mandibular Condylar Cartilage

Deparaffinized histological sections were used for hematoxylin and eosin (H-E) staining and Alcian Blue staining. Briefly, H-E staining was performed using hematoxylin and eosin dyes (Fujifilm Wako chemicals), and Alcian Blue staining was performed using an Alcian Blue staining kit (Bio Future Technologies Inc., Tokyo, Japan), according to the manufacturer’s instructions. Stained sections were used for the measurement of the thickness of the mandibular condylar cartilage. The thickness of each layer was measured using Image J software version 1.52 (National Institutes of Health, Bethesda, MD, USA). We divided the mandibular condylar cartilage into three layers: fibrous layer, proliferative cell layer, and maturative cell layer. The fibrous layer consists of fibrous connective tissue, which protects the underlying layers. The proliferative cell layer is composed of polygonal-shaped cells with faint cytoplasm. The maturative cell layer is characteristic of chondrocytes [40]. 

### 2.5. Immunohistochemical Staining for Pik3cd

The deparaffinized histological sections were quenched for endogenous peroxidase activity by incubation with 0.3% H_2_O_2_ in methanol for 30 min; then the sections were treated with Block Ace (DS Pharma Biomedical Inc., Osaka, Japan) for 1 h. The sections were incubated with anti-Pik3cd monoclonal antibody (150-fold dilution; ma5-26520, Thermo Fisher Scientific, Waltham, MA, USA) in Can Get Signal^®^ immunostain immunoreaction enhancer solution (Toyobo Co., Ltd., Osaka, Japan) for 2 h at room temperature. After washing with PBS containing 0.5% Tween 20 (PBS-T), the sections were incubated with peroxidase-conjugated secondary antibody (1000-fold dilution; A90-216P, Fortis Life Sciences, Waltham, MA, USA) for 2 h at room temperature, then thoroughly washed with PBS-T. Then, sections were flooded with DAB solution (Vector Laboratories, Burlingame, CA, USA) and incubated for 1 min at room temperature. Photographs were taken with a microscope (BZ-9000; Keyence Co., Osaka, Japan). 

The percent of stained area per field was calculated by using Image J software from 9 images of both mandibular condylar cartilage and femur articular cartilage.

### 2.6. Statistical Analysis

Data are expressed as the mean ± standard deviation (SD) or mean ± standard error of the mean (SEM). Comparisons were performed using the Student’s *t*-test by Microsoft Excel (Microsoft Corp., Redmond, WA, USA). A *p* < 0.05 was considered statistically significant.

## 3. Results

### 3.1. Myo-Inositol Had No Effect on Body Weight

Firstly, time-course changes in body weight in both groups were examined to determine whether myo-inositol supplementation influences systemically or not (Figure 1). As 6 weeks old is just after weaning, the mean body weight of both groups increased with age until they were 25 weeks old. Both groups seemed to reach a plateau and maintained their body weight until the end of the experiment. No statistically significant difference in body weight at each time point between the groups was observed during the whole experimental period.

This result suggests that myo-inositol does not have a promotional effect on systemic body growth.

### 3.2. Myo-Inositol Had No Effect on Maxillary Length and Femur Length

We then analyzed micro CT data of the maxillary and femur to clarify whether myo-inositol had an influence systemically (Figure 2). The maxillary length of the control group was 45.0 ± 1.5 mm, and that of the myo-inositol group was 43.4 ± 1.6 mm, and no statistically significant difference between the groups was observed. As to the femur length, the control group was 91.3 ± 0.9 mm, and the myo-inositol group was 90.6 ± 1.1 mm. There was no statistically significant difference between the groups in femur length. This result suggests that myo-inositol does not have any promotional effect on bone growth in the maxilla and femur.

### 3.3. Myo-Inositol Specifically Augmented the Mandibular Growth

Next, we measured the mandibular length by micro CT and found that the average length of the control and myo-inositol groups was 74.95 ± 0.53 mm and 78.40 ± 0.59 mm, respectively (Figure 3). There were statistically significant differences between the groups. There was about 4.7% augmentation of mandibular length by myo-inositol. This result suggests that myo-inositol has specific augmentation of mandibular growth.

### 3.4. Pik3cd Is Strongly Expressed in Mandibular Condylar Cartilage

Previously, we discovered that Pik3cd is specifically strongly expressed in mandibular condylar cartilage in mice [34], which gives specific augmentation of mandibular growth by myo-inositol. Therefore, the expression of Pik3cd in rabbit cartilage tissues was examined. Immunohistological staining clearly demonstrated that mandibular condylar cartilage strongly expresses Pik3cd as compared to femur articular cartilage (Figure 4a–d). There was no difference in the extent of Pik3cd expression in the same cartilage between the groups; in other words, mandibular condylar cartilage exhibited strong Pik3cd expression as compared to femur articular cartilage in both groups. The calculated percent area of pik3cd positive in mandibular condylar cartilage was 8.3 ± 0.5, and that of femur articular cartilage was 0.4 ± 0.1. There was a statistically significant difference between mandibular condylar cartilage and femur articular cartilage. This result suggests that Pik3cd was specifically strongly expressed in mandibular condylar cartilage, even in rabbits.

### 3.5. Myo-Inositol Increased the Thickness of Mandibular Condylar Cartilage

We further examined the augmentation of mandibular growth histologically (Figure 5). The images of H-E staining clearly recognize the surface fibrous layer and the underlying proliferative cell layer, though it was relatively difficult to separate the proliferative cell layer and the underlying maturative chondrocyte layer (Figure 5a,b). On the other hand, the images of Alcian Blue staining clearly recognize the proliferative cell layer and underlying maturative chondrocyte layer (Figure 5c,d). In both stainings, the maturative chondrocyte layer seemed to be thicker in the myo-inositol group as compared to that of the control group.

We then measured the thickness of each cell layer between the groups using both images of H-E staining and Alcian Blue staining (Figure 5e,f). The thickness of whole layers was statistically significantly thicker in the myo-inositol group (356.5 ± 8.2 μm in H-E staining and 336.0 ± 14.4 μm in Alcian Blue staining) as compared to that in the control group (278.8 ± 5.1 μm in H-E staining and 287.2 ± 6.8 μm in Alcian Blue staining) in both images. There was no statistical difference in the thickness of the fibrous layer between the groups (H-E staining: 209.8 ± 2.4 μm in the control group and 209.8 ± 6.9 μm in the myo-inositol group; Alcian Blue staining: 209.7 ± 7.6 μm in the control group and 209.6 ± 9.1 μm in the myo-inositol group). The thickness of the maturative chondrocyte layer was thicker in the myo-inositol group (169.0 ± 5.2 μm in H-E staining and 157.4 ± 6.6 μm in Alcian Blue staining) as compared to that in the control group (108.6 ± 4.2 μm in H-E staining and 118.8 ± 3.5 μm in Alcian Blue staining). Interestingly, the proliferative cell layer was thicker in the control group (39.6 ± 1.1 μm in H-E staining and 41.3 ± 1.5 μm in Alcian Blue staining) than in the myo-inositol group (30.8 ± 1.0 μm in H-E staining and 30.9 ± 1.9 μm in Alcian Blue staining). These results suggest that myo-inositol would increase the maturative cell layer, which results in the increase of the whole layer of mandibular condylar cartilage.

## 4. Discussion

In this manuscript, we discovered that the nutritional supplementation of myo-inositol in laboratory chow during the growth period specifically augmented mandibular growth without any systemic influence, not only in mice [34] but also in rabbits. Myo-inositol is one of the sugar alcohols, with a molecular weight of 180.16, and is known to play a role as a number of secondary messengers in eukaryotic cells [41]. Myo-inositol is naturally present in a variety of foods, including fruits, beans, grains, and nuts [42]. Our results indicated that growth augmentation of the mandible with myo-inositol requires no local injection but just simply needs to supplement the food. As to the specificity of growth augmentation in mandibular condylar cartilage by myo-inositol, Pik3cd expression was extensive in mandibular condylar cartilage as compared to femur articular cartilage. This site-specific Pik3cd expression, similar to the results of mice [34], would play a role in the myo-inositol-mediated sole augmentation of mandibular growth. Our results suggest the possibility of internal orthodontic treatment, which would check and supplement myo-inositol for the growing mandibular retrognathism patients. 

As to the extent of growth augmentation in the mandible by supplementation of myo-inositol, it was 4.7% induction in our experiments using rabbits. Our previous experiment revealed 8.4% induction in mice [34]. Comparing the percentage of induction in both animals, the induction in rabbits was relatively smaller than in mice despite the fact that the concentration of myo-inositol in laboratory chow was the same in both experiments. One possible explanation for this reduction in rabbits may be the difference in the amount of food intake. Though we provided free access to water and food throughout the experiment for both animals, experimental animals sometimes played with food, which reduced the actual food intake [43]. Detailed estimation of actual food intake should be monitored in future experiments. Another explanation might be the difference in the reactivity to myo-inositol. There was no report describing the comparison of absorption efficiency of myo-inositol from the intestinal tract, which affects the difference in serum myo-inositol concentration between the animals.

Review articles summarize the mean increase in mandibular length by functional appliances such as Herbst and Sander Bite Jumping. Adriana et al. reported that the Sander Bite Jumping reported the greatest increase in mandibular length, 3.4 mm, followed by Twin Block, Bionator, Harvold Activator, and Frankel devices [44]. Stefanos et al. reported that the Herbst appliance gave a greater 1.5 mm increase in mandibular length [45]. Ra’ed et al. compared both skeletal and dental effects of the Herbst appliance anchored with temporary anchorage devices and the Herbst appliance alone and found that the Herbst appliance anchored with temporary anchorage devices exhibited longer mandibular length by a mean difference of 3.7 mm as compared to the Herbst appliance alone. Considering the mean mandibular length (Condylion to Gnathion) in Japanese adult males is 125.5 mm [46], the percentage of above mandibular growth augmentation would be 2.7, 1.2, and 2.9%, respectively. Therefore, mandibular growth augmentation by nutritional supplementation of myo-inositol is superior to functional appliance. Our results were obtained in animal experiments; further validation in clinical research is mandatory in the future.

As to the effects of functional appliances, not only skeletal [10,11] but also dentoalveolar effects [12] are reported to be pivotal. Furthermore, a functional improvement from the point of the airway in relation to OSA was also reported [47]. Therefore, the extent of effectiveness in the correction of Class II malocclusion is dependent on the parameter for evaluation. Improvement of Class II malocclusion was achieved and judged by the value of Wits appraisal [13], which evaluates the intermaxillary relationship based on the occlusal plane [14]. However, the improvement of Class II malocclusion was relatively weak, judged by Pogonion to N-perpendicular, which indicates the relationship between the chin and cranial base [13]. The mandibular growth augmentation by myo-inositol was evaluated by the length of the mandible, which solely depends on the mandibular growth. Therefore, augmentation of mandibular growth by myo-inositol would be useful from the point of the skeletal effects.

Regarding the growth potential of the mandible, previous research demonstrated that mastication [48,49,50], functional interference between maxillary and mandibular dental arch [51,52,53,54,55], and growth potential at mandibular condylar cartilage [56,57,58] play pivotal roles. Some orthodontists use functional appliances with clenching training and obtain some additive growth augmentation for functional appliance therapy [59]. Therefore, a combination of these parameters, such as mastication and growth potential simultaneously with myo-inositol application, has a chance to augment mandibular growth more effectively. Prediction of mandibular growth prior to the treatment is mandatory for orthodontists, and there are many methods for prediction, such as the Jarabak method [60].

There are some reported nutritional factors that augment chondrocytic growth and differentiation. Lipophilic vitamins such as Vitamins A, D, E, and K [29], glucosamine and chondroitin sulfate [30], glucose and glucose-derived sugars [31], and vitamin D [32] were reported to have positive effects on chondrocytic growth and differentiation. Interestingly, a potential role for vitamin D in sleep arousals was reported [61]. These factors exhibit non-specific augmentation of cartilage and chondrocyte growth. Therefore, specific mandibular growth augmentation with these factors is quite difficult to obtain. Local injections are necessary for mandibular-specific growth augmentation in case of the use of growth factors [28,62,63]. Considering this local injection, nutritional supplementation of myo-inositol has an advantage compared to the other growth factors from the point of safety perspective. Growth hormone therapy has been indicated to increase height in children with short stature. However, this method has various drawbacks, including the need for frequent injections and the positive correlation between IGF-1 blood levels and the incidence of prostate, breast, and colorectal cancers [64].

This study has some limitations. Regarding the role of Pik3cd in myo-inositol-mediated growth augmentation, we did not perform any inhibition experiments against Pik3cd. However, our previous report clearly demonstrates that the inhibition of Pik3cd by a chemical inhibitor almost completely inhibited myo-inositol-mediated augmentation of chondrocyte proliferation in mice [34]. Together, Pik3cd would play a role in myo-inositol-mediated growth augmentation, even in rabbits. Further confirmatory experiments are necessary on whether Pik3cd plays a role in myo-inositol-mediated growth augmentation, even in rabbits.

Histological examination revealed that myo-inositol increased the thickness of mandibular condylar cartilage, especially the thickness of the maturative cell layer. Coincidentally, Pik3cd expression was intense in the maturative cell layer, which makes sense that nutritional supplementation of myo-inositol augments cellular proliferation and cartilage differentiation. As we previously discovered that myo-inositol augments chondrocytic differentiation [37] and proliferation [34] in mice, our results confirmed that nutritional supplementation of myo-inositol in laboratory chow during growth period specifically augmented mandibular growth by the increase of the maturative cell layer of mandibular condylar cartilage. Mandibular condylar cartilage is generally divided into five layers: fibrous layer, proliferative cell layer, transitional cell layer, maturative cell layer, and hypertrophic cell layer [65]. In this present study, we divided it into three layers, fibrous layer, proliferative cell layer, and maturative cell layer. The fibrous layer consists of fibrous connective tissue, which protects the underlying layers; the proliferative cell layer is composed of polygonal-shaped cells with faint cytoplasm; and the maturative cell layer is characteristic of chondrocytes [40]. These layers exhibit different cell statuses, such as gene expression profiles [66], which would make different reactivity against various factors, including myo-inositol. 

As to the pharmacological use of myo-inositol against diseases, there are several established targets. Dietary supplements for polycystic ovary syndrome are one of the common pharmacological uses of myo-inositol [67,68]. Myo-inositol is also used for bipolar disorder [69] and gestational diabetes [70]. Besides these pharmacological uses, myo-inositol is thought to be a higher-safety substance. Sixty-five substances have been designated by the Minister of Health, Labor and Welfare, Japan, as substances that are clearly not hazardous to human health. Myo-inositol is included in the list. In addition, the review article indicated that even the highest dose of myo-inositol (12 g/day) induced only mild gastrointestinal side effects such as nausea, flatus, and diarrhea, which signified higher safety of myo-inositol [71]. As we confirmed specific mandibular growth by nutritional supplementation of myo-inositol in mice and rabbits experiments, a clinical test is necessary to clarify the effectiveness of myo-inositol in humans. Regarding the safety test of myo-inositol in humans, since myo-inositol is already being used as a treatment for other diseases, it is assumed that there is no need to conduct new safety studies.

Figure 6 summarizes our proposed orthodontic treatment for the patient with mandibular retrognathism during the growth period in the near future. At present, orthodontists use functional appliances such as the Bionator or Activator for the treatment of mandibular retrognathism by the augmentation of mandibular growth, though the effect is lowly predictable, especially in the long term. In the future, nutritional supplementation of myo-inositol would be a potential therapeutic treatment for patients with mandibular retrognathism during the growth period. Orthodontists would check not only inter-maxillary relationships but also monitor serum myo-inositol levels during the growth period and supplement myo-inositol in case the patients exhibit low levels of myo-inositol.

Together, skeletal Class II is not just the problem of intermaxillary relationships between the maxilla and mandible but exhibits functional problems, including sleep [8] and temporomandibular disorders [4,5], which severely reduce quality of life [72,73]. Therefore, prevention or improvement of skeletal Class II during the growing stage would be highly beneficial for the patients.

## 5. Conclusions

In conclusion, we discovered that the nutritional supplementation of myo-inositol during the growth period specifically augmented mandibular growth without any systemic influence not only in mice [34] but also even in rabbits. Our results suggest the possibility of clinical use of myo-inositol for augmentation of the mandibular growth in growing mandibular retrognathism patients in the future.

## Figures and Tables

**Figure 1 dentistry-12-00049-f001:**
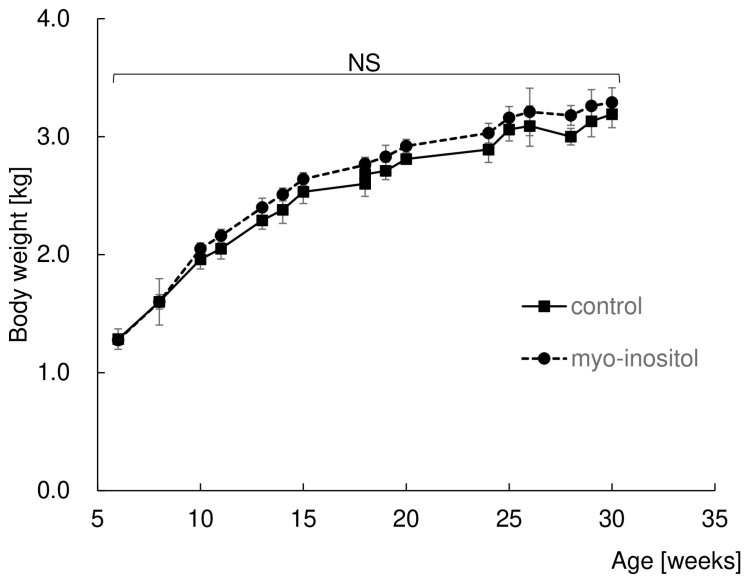
Myo-inositol had no effect on body weight. The change in body weight during the experiment was monitored every week. The mean body weight of the control group is shown by a square and solid line, and the myo-inositol group is shown by a circle and dotted line. Results are expressed as mean ± SD. There was no statistically significant difference between the groups. NS: no significant difference between the groups.

**Figure 2 dentistry-12-00049-f002:**
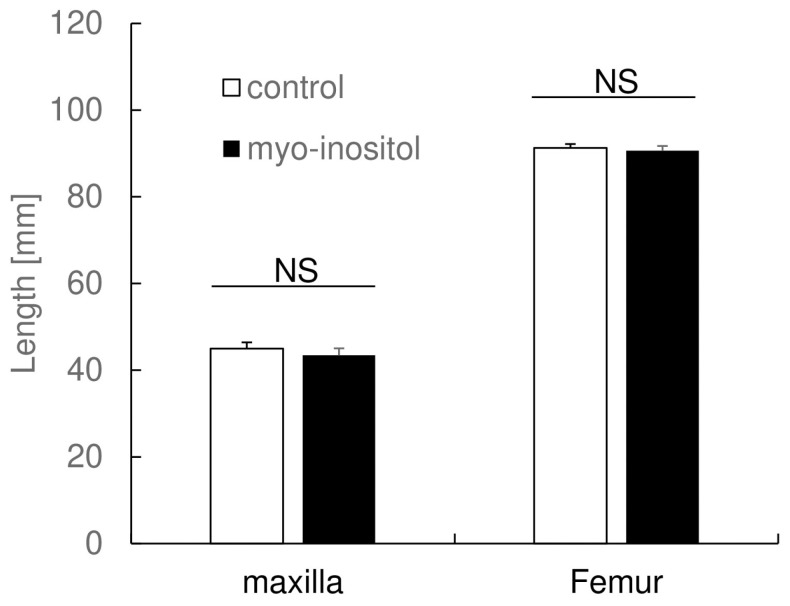
Myo-inositol had no effect on maxillary length and femur length. The maxillary and femur lengths in each group were measured using micro CT images. Results are expressed as mean ± SD. Open bars represent the control group, and closed bars represent the myo-inositol group. NS: no significant difference between the groups.

**Figure 3 dentistry-12-00049-f003:**
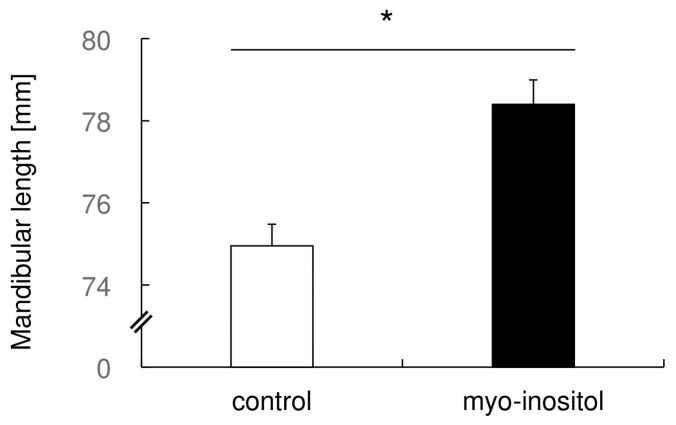
Myo-inositol specifically augmented the mandibular growth. The length of the mandible in each group was measured using micro CT images. Results are expressed as mean ± SD. Open bars represent the control group, and closed bars represent the myo-inositol group. *: *p* < 0.05 between the groups.

**Figure 4 dentistry-12-00049-f004:**
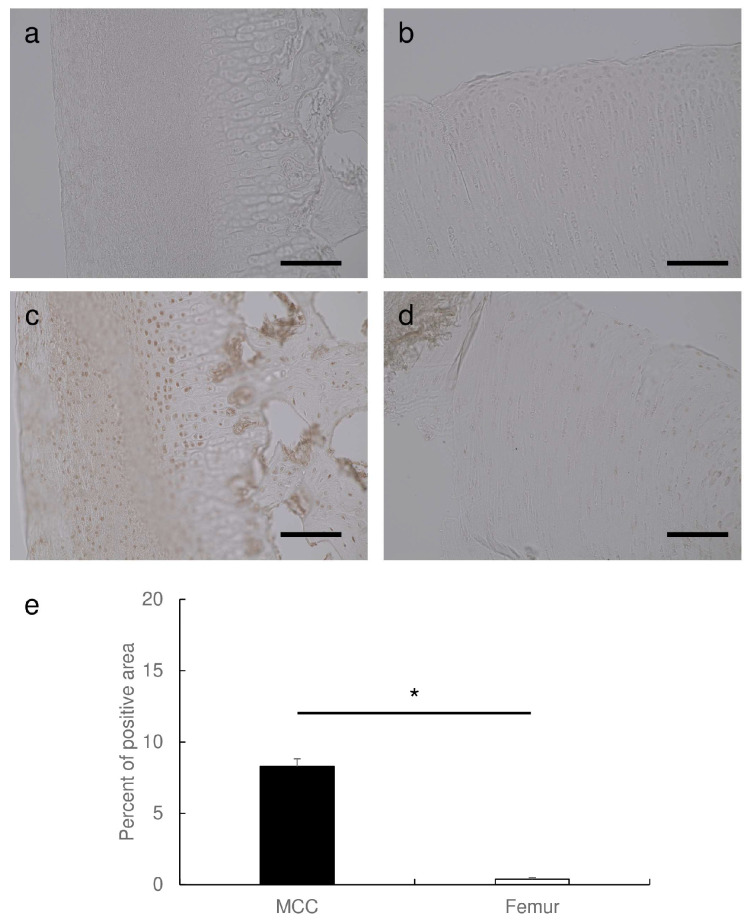
Pik3cd is strongly expressed in mandibular condylar cartilage. Representative images of immunohistochemical staining for Pik3cd taken under an ×20 objective lens are shown. (**a**,**b**) exhibits background level staining by using only secondary antibody, (**c**,**d**) exhibits the positive staining for Pik3cd by using primary and secondary antibodies, (**a**,**c**) are the sections of mandibular condylar cartilage, and (**b**,**d**) are the sections of femur articular cartilage. Bar: 100 μm. (**e**) indicates the percent positive area per field in mandibular condylar cartilage (MCC) and femur articular cartilage. Results are expressed as mean ± SEM. *: *p* < 0.05.

**Figure 5 dentistry-12-00049-f005:**
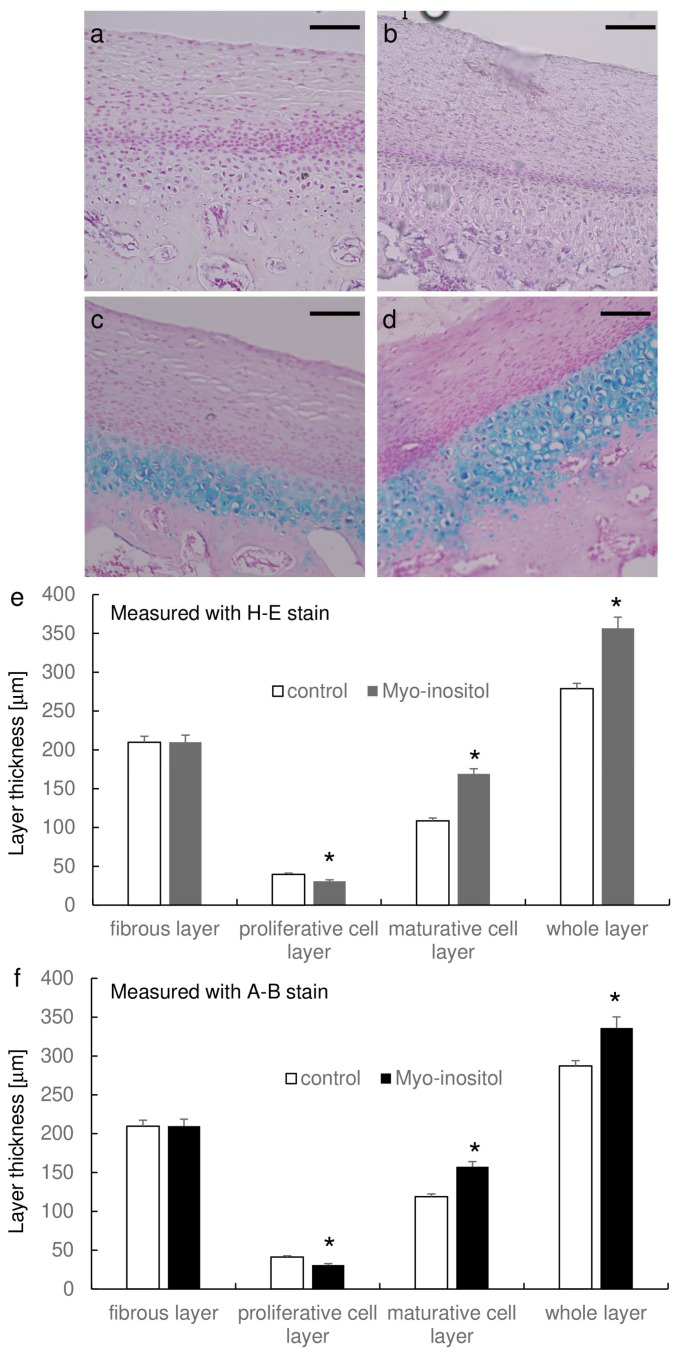
Myo-inositol increased the thickness of mandibular condylar cartilage. Representative images of H-E staining (**a**,**b**) and Alcian blue staining (**c**,**d**) are shown. (**a**,**b**) are the sections of the control group, and (**b**,**d**) are the sections of the myo-inositol group. Bar: 100 μm. (**e**,**f**) represents the mean layer thickness of the groups. (**e**) shows the thickness measured with the H-E stain, and (**f**) shows the thickness measured with Alcian blue staining. Results are expressed as mean ± SEM. *: *p* < 0.05 between the groups.

**Figure 6 dentistry-12-00049-f006:**
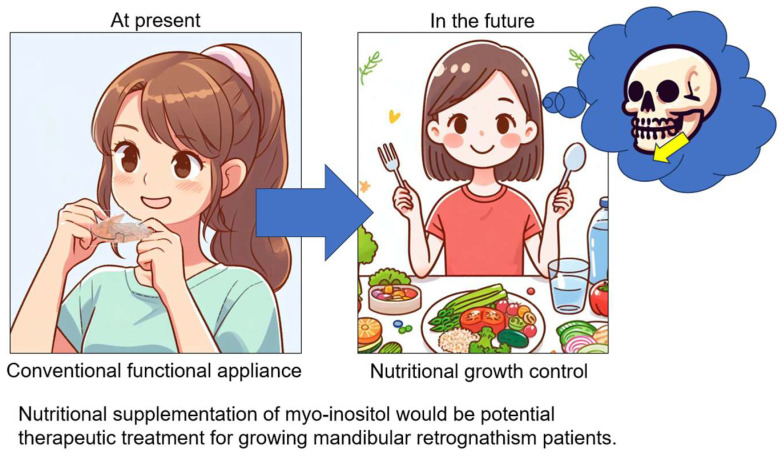
Schematic illustration of our proposed orthodontic treatment. At present, orthodontists use functional appliances such as the Bionator or Activator for the treatment of mandibular retrognathism by the augmentation of mandibular growth, though the effect is lowly predictable, especially in the long term. In the future, nutritional supplementation of myo-inositol would be a potential therapeutic treatment for patients with mandibular retrognathism during the growth period.

## Data Availability

The datasets used and/or analyzed during the current study are available from the corresponding author upon reasonable request.

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
