# Peer review of "Mandibular Endochondral Growth Is Specifically Augmented by Nutritional Supplementation with Myo-Inositol Even in Rabbits"

_dentistry, 2024, doi:10.3390/dj12030049_

Round 1

Reviewer 1 Report

Comments and Suggestions for Authors

The manuscript “Specific mandibular endochondral growth augmentation by nutritional supplementation with myo-inositol would be alternative orthopedic treatment modality to functional appliances“ investigates myo-inositol supplementation in japanese white rabbits. 5 rabbits were fed with a normal diet and 5 rabbits were fed with a diet high in myo-inositol. Mandibular growth was investigated using micro-CT and histological slides. The conclusion that can be drawn from the results is that myo-inositol increases mandibular growth in rabbits.

Scientific research on this condition is appreciated. The manuscript is generally well structured and adds to the literature on mandibular growth modification in animals.

My main point of criticism is the generalizability/translation of the finding of this animal study. The conclusions of the study regarding the clinical use of myo-inositol in orthodontic patients should be weakened.

The following issues should be addressed:

In my opinion, the title is too offensive. We do not yet know if inositol is an alternative to functional appliances in clinical orthodontics.

Abstract: Please simplify and write more concisely according to the journal guidelines: “The abstract should be a single paragraph and should follow the style of structured abstracts, but without headings”. The last sentence does exaggerate the main conclusion.

Please explain why a reporting guideline (e. g. ARRIVE) was not used.

Sample size: Although the sample size is mentioned (n=5 for each group), it would be beneficial to explain how this sample size was determined and if any power analysis was conducted.

Inclusion/exclusion criteria: Please describe any criteria used for including and excluding animals. Were animals excluded from the analysis?

Baseline characteristics: Please specify the baseline animal characteristics. Was the selection of six-week-old male Japanese white rabbits based on a scientific rationale?

Housing conditions: Please provide more details on the housing conditions.

Randomization: Were any efforts made to randomize the assignment of rabbits to the control and myo-inositol groups?

Blinding: Were the individuals conducting the micro-CT analysis blinded to the group assignments?

Conclusions: The conclusion should be more concise. The conclusions of the study on the clinical use of myo-inositol in orthodontic patients should be weakened. Figure 6 is highly controversial and should be part of the discussion.

I hope that these points will help to improve the article and look forward to receiving the revised manuscript.

Comments on the Quality of English Language

Moderate editing of English language required.

Author Response

Thank you very much for fruitful discussion and suggestion. We revised our manuscript according to the reviewers’ comments, and detailed response are written below.

Reviewer-1

  1. “The manuscript “Specific mandibular endochondral growth augmentation by nutritional supplementation with myo-inositol would be alternative orthopedic treatment modality to functional appliances“ investigates myo-inositol supplementation in Japanese white rabbits. 5 rabbits were fed with a normal diet and 5 rabbits were fed with a diet high in myo-inositol. Mandibular growth was investigated using micro-CT and histological slides. The conclusion that can be drawn from the results is that myo-inositol increases mandibular growth in rabbits. Scientific research on this condition is appreciated. The manuscript is generally well structured and adds to the literature on mandibular growth modification in animals.”

---Response:       Thank you for your insightful comments and for your interest in our manuscript.

  1. “My main point of criticism is the generalizability/translation of the finding of this animal study. The conclusions of the study regarding the clinical use of myo-inositol in orthodontic patients should be weakened.” “The following issues should be addressed: In my opinion, the title is too offensive. We do not yet know if inositol is an alternative to functional appliances in clinical orthodontics.”

---Response:       As suggested, we modified the conclusion and title not to be overstatement.

  1. “Abstract: Please simplify and write more concisely according to the journal guidelines: “The abstract should be a single paragraph and should follow the style of structured abstracts, but without headings”. The last sentence does exaggerate the main conclusion.”

---Response:       Thank you very much for kind suggestion. We modified abstract accordingly.

  1. “Please explain why a reporting guideline (e. g. ARRIVE) was not used.”

---Response:       Thanks for this important suggestion. We filled ARRIVE guideline form and modified main text to add required information.

  1. “Sample size: Although the sample size is mentioned (n=5 for each group), it would be beneficial to explain how this sample size was determined and if any power analysis was conducted.”

---Response:       As suggested, we added the sentence explaining how we estimate sample number in the materials and methods section. Briefly, we performed power analysis to estimate required number of samples in each group. For power analysis, effect size, alpha error probability, power (1 – beta error probability) are required for estimation. As we previously reported the growth augmentation effect of myo-inositol (Bone 2019, 121, 181-190), we calculated the value of effect size using the data for previous report. Mean and SD of the control and experimental groups in the previous report were 8.82 ± 0.106 and 9.64 ± 0.435, respectively, which gave effect size value as 2.58. Power analysis in the condition of effect size = 2.58, alpha error probability = 0.05, power = 0.8, gave required sample size as 4. We set n = 5 in each group in this experiment, which clear the required sample size.

  1. “Inclusion/exclusion criteria: Please describe any criteria used for including and excluding animals. Were animals excluded from the analysis?”

---Response:       Thank you very much for precious comments relate to inclusion and exclusion criteria of subjects. As we conducted the experiments using experimental animals, uniformity of each experimental animal is quite high as compared to that of clinical research subjects. Therefore, we purchased the experimental animals, pooled before grouping to maintain uniformity of the animals between the groups (randomization), and used all animals for analysis. Fortunately, there was no experimental animal which exhibits severe symptoms and death during the experimental period.

  1. “Baseline characteristics: Please specify the baseline animal characteristics. Was the selection of six-week-old male Japanese white rabbits based on a scientific rationale?”

---Response:       Thank you for your insightful comments regarding how chose experimental animals as rabbits. As to the larger experimental animals over mice would be rats, rabbits, dogs, pigs, and monkeys. Among these possible experimental animals, our facility for experimental animals allows rats, rabbits, and dogs. Comparing the size of the experimental animals, rats are not so large as compared to rabbits and dogs. Therefore, we did not use rats in this experiment. Then we considered which animal is better to our experiment, rabbits or dogs. As the genetic homology is relatively different in dog against mice, as compared to that in rabbits and mice, the difficulty in genetic analysis or protein level analysis using antibody would be suspected of the analysis in dogs. Therefore, we chose rabbits as the experimental animal in this time.

As to the age of the experimental rabbits, weaning of rabbits is generally 3 to 4 weeks old, and 1 week acclimation period was required prior to experiment. Therefore, we started the experiment at six-week-old.

  1. “Housing conditions: Please provide more details on the housing conditions.”

---Response:       Thank you very much for precious comments. The details of housing conditions for experimental animals were added.

  1. “Randomization: Were any efforts made to randomize the assignment of rabbits to the control and myo-inositol groups?”

---Response:       Thank you very much for indicating the point. As described in response 6, the experimental animals were firstly pooled before grouping to maintain uniformity of the animals between the groups (randomization) and used all animals for analysis.

  1. “Blinding: Were the individuals conducting the micro-CT analysis blinded to the group assignments?”

---Response:       Regarding the blinding during analysis, we set unique name of each dicom folder for each animal to make blinding at the stage of converting microCT raw data to dicom files. This step made it possible to analyze dicom data with blinding. We added the sentence in the materials and methods section relate to this issue.

  1. “Conclusions: The conclusion should be more concise. The conclusions of the study on the clinical use of myo-inositol in orthodontic patients should be weakened. Figure 6 is highly controversial and should be part of the discussion.

---Response:       Thank you very much for kind suggestion. We modified conclusion and discussion accordingly. As this manuscript is submitted to special issue, “Orthodontics and New Technologies” in the Dentistry Journal, we thought it is better to show some future direction of orthodontic therapy even if it is still under research level. Therefore, we summarized possibilities of our present research in the figure 6. We moved figure 6 to the discussion, not in the conclusion, according to the reviewer’s suggestion.

  1. “I hope that these points will help to improve the article and look forward to receiving the revised manuscript.”

---Response:       Thank you very much for fruitful suggestions to improve our manuscript. We wish the revised manuscript meets the standards expected for publication.

Reviewer 2 Report

Comments and Suggestions for Authors

The present research demonstrated how supplementation with myo-inositol boosts mandible growth in rabbits. Generally speaking, It showed some novelty and scientific soundness, except for some concerns:

1. the relatively small group size may be a problem--only 5 animals in each group. Are there any considerations or calculations that prove 5 rabbits sufficient?

2. maybe there could be more than two groups of rabbits. adding the myo-inositol-free groups and different amounts of myo-inositol supplement groups may better depict the dose-dependent effect between myo-inositol and mandible growth.

3. all rabbits were raised in a masticatory-free condition, which is uncommon and may inhibit mandible growth. Can the present study's result be replicated in rabbits with normal chewing?

Author Response

  1. “The present research demonstrated how supplementation with myo-inositol boosts mandible growth in rabbits. Generally speaking, it showed some novelty and scientific soundness, except for some concerns:”

---Response:       Thank you for your insightful comments and for your interest in our manuscript.

  1. “The relatively small group size may be a problem--only 5 animals in each group. Are there any considerations or calculations that prove 5 rabbits sufficient?”

---Response:       As suggested, we added the sentence explaining how we estimate sample number in the materials and methods section. Briefly, we performed power analysis to estimate required number of samples in each group. For power analysis, effect size, alpha error probability, power (1 – beta error probability) are required for estimation. As we previously reported the growth augmentation effect of myo-inositol (Bone 2019, 121, 181-190), we calculated the value of effect size using the data for previous report. Mean and SD of the control and experimental groups in the previous report were 8.82 ± 0.106 and 9.64 ± 0.435, respectively, which gave effect size value as 2.58. Power analysis in the condition of effect size = 2.58, alpha error probability = 0.05, power = 0.8, gave required sample size as 4. We set n = 5 in each group in this experiment, which clear the required sample size.

  1. “maybe there could be more than two groups of rabbits. adding the myo-inositol-free groups and different amounts of myo-inositol supplement groups may better depict the dose-dependent effect between myo-inositol and mandible growth.”

---Response:       Thank you very much for precious comments. In this report, we set only two myo-inositol contents, 9.7 mg/kg for control group and 66 g (66000mg)/kg for experimental group. Ideally speaking, it is better to make three groups, similar to previous report performed using experimental mice (Bone 2019, 121, 181-190). In the previous paper, we set control group (myo-inositol content: 9.7 mg/kg), the low concentration group (myo-inositol content: 6.6 g/kg), and the high concentration group (myo-inositol content: 66 g/kg). We primarily wanted to know whether rabbits also exhibit myo-inositol-mediated specific mandibular growth augmentation similar to mice, we only set two groups, control and experimental groups. As the reviewer suggested, it is mandatory to clarify the dose-response of myo-inositol-mediated mandibular growth augmentation for future clinical use in human. Therefore, we are now planning to perform additional experiments to clarify the issue in the near future.

  1. all rabbits were raised in a masticatory-free condition, which is uncommon and may inhibit mandible growth. Can the present study's result be replicated in rabbits with normal chewing?

---Response:      Thank you very much for indicating the point. Regarding the effect of mastication on mandibular growth, we cited two references, “To eliminate possible influence of mastication of the laboratory chew on mandibular growth [37,38]” in the materials and methods section.

  1. Enomoto, A.; Watahiki, J.; Yamaguchi, T.; Irie, T.; Tachikawa, T.; Maki, K. Effects of mastication on mandibular growth evaluated by microcomputed tomography. European journal of orthodontics 2010, 32, 66-70.
  2. Hichijo, N.; Kawai, N.; Mori, H.; Sano, R.; Ohnuki, Y.; Okumura, S.; Langenbach, G.E.; Tanaka, E. Effects of the masticatory demand on the rat mandibular development. Journal of oral rehabilitation 2014, 41, 581-587.

Reference 37 revealed some difference in mandibular length between soft food and hard food groups. On the other hand, reference 38 reported no statistical differences between soft food and hard food groups. These contradictory reports signify that mastication gives some influence on mandibular length Under certain circumstances. In addition, both reports revealed hard food group exhibited longer ramus height, which means mastication induces growth at gonion. Furthermore, clenching training gives some additive growth augmentation for functional appliance therapy (Orthodontic waves 2005, 64, 106-113). Taken together, it is mandatory to clarify the effect of myo-inositol on mandibular growth under mastication. We are now planning to perform additional experiments to clarify the issue in the near future. In addition, we added the description of the relationship between mandibular growth and mastication in the discussion.

Reviewer 3 Report

Comments and Suggestions for Authors

Dear Authors, thank you for this interesting article. 

Here are some suggestions of mine:

1. The first paragraph describes the Class II with retrognathism and refers to some conditions that might be present. Please, widen that, especially the aspect of TMD and OSA - use the abbrevations and explain them in this place. The topic of OSA is in the eye of interests of many researchers lately and therefore I think it should be described more detailly - refer to Sleep architecture and vitamin D in hypertensives with obstructive sleep apnea, but also to correlation between the airway volume and the hyoid bone position, types of malocclusion and palatal depth

2. Line 47 (and further) - give referenece number directly after the name of the Author.

3. In the discussion, you should focus on "why" do the activators or other functional appliances change the mandibular lenght - it is known that widening of maxilla "unblocks" the mandible and lets it grow, so probably it is not caused only because of the activator, but also due to the fact, that the mandible is "allowed" to grow. Jarabak described detaillly the cephalometric analysis that makes it possible to "predict" the type and the amount of growth of the indivisuals, which also should be added to the discussion (search for brachyfacial and oligofacial types of faces and growth patterns)

4. Line 318-320, please refer also to the vit D3 and OSA in the discussion, see:

Kanclerska J, Wieckiewicz M, Nowacki D, Szymanska-Chabowska A, Poreba R, Mazur G, Martynowicz H. Sleep architecture and vitamin D in hypertensives with obstructive sleep apnea: A polysomnographic study. Dent Med Probl. 2023 Oct 20. doi: 10.17219/dmp/172243.

and the CLass II malocclusion and influence of it on temporomandibular disordes, eg. 

Tahmasbi S, Seifi M, Soleymani AA, Mohamadian F, Alam M. Comparative study of changes in the airway dimensions following the treatment of Class II malocclusion patients with the twin-block and Seifi appliances. Dent Med Probl. 2023;60(2):247–254. doi:10.17219/dmp/142292

5. I would add the separate subchapter of "limitations" and "targets", as the Authors point them strongly out and therefore I think the Reader should pay careful attention to it. 

6.In the end of the discussion, you should also add the aspect of "Relationship between pain severity, satisfaction with life and the quality of sleep.. (especially in patients with temporomandibular disorders), focusing on the multifactorial aspects of TMD and OSA - this is not only bite, occlusion, but also many factors and this should be at least one paragraph in the discussion.

I rate the paper higyh, although some corrections should be done Thank you.

Author Response

  1. “Dear Authors, thank you for this interesting article.  Here are some suggestions of mine:”

---Response:       Thank you for your insightful comments and for your interest in our manuscript.

  1. “The first paragraph describes the Class II with retrognathism and refers to some conditions that might be present. Please, widen that, especially the aspect of TMD and OSA - use the abbreviations and explain them in this place. The topic of OSA is in the eye of interests of many researchers lately and therefore I think it should be described more detailly - refer to Sleep architecture and vitamin D in hypertensives with obstructive sleep apnea, but also to correlation between the airway volume and the hyoid bone position, types of malocclusion and palatal depth”

---Response:       Thank you very much for this important suggestion. As suggested, we deepen the topics related to OSA in the introduction.

  1. Line 47 (and further) - give reference number directly after the name of the Author.

---Response:       As suggested, we have corrected the position of the reference number in the text.

  1. In the discussion, you should focus on "why" do the activators or other functional appliances change the mandibular length - it is known that widening of maxilla "unblocks" the mandible and lets it grow, so probably it is not caused only because of the activator, but also due to the fact, that the mandible is "allowed" to grow. Jarabak described detailly the cephalometric analysis that makes it possible to "predict" the type and the amount of growth of the individuals, which also should be added to the discussion (search for brachyfacial and oligofacial types of faces and growth patterns)

---Response:       As suggested, we have corrected the position of the reference number in the text.

  1. Line 318-320, please refer also to the vit D3 and OSA in the discussion, see: Kanclerska J, Wieckiewicz M, Nowacki D, Szymanska-Chabowska A, Poreba R, Mazur G, Martynowicz H. Sleep architecture and vitamin D in hypertensives with obstructive sleep apnea: A polysomnographic study. Dent Med Probl. 2023 Oct 20. doi: 10.17219/dmp/172243. and the CLass II malocclusion and influence of it on temporomandibular disordes, eg.  Tahmasbi S, Seifi M, Soleymani AA, Mohamadian F, Alam M. Comparative study of changes in the airway dimensions following the treatment of Class II malocclusion patients with the twin-block and Seifi appliances. Dent Med Probl. 2023;60(2):247–254. doi:10.17219/dmp/142292

---Response:       Thank you very much for precious suggestion and giving us the key references. As suggested, we modified 5th paragraph of the discussion accordingly, with use of former reference. Latter suggested reference was used in the 4th paragraph of the discussion.

  1. I would add the separate subchapter of "limitations" and "targets", as the Authors point them strongly out and therefore, I think the Reader should pay careful attention to it. 

---Response:      Thank you very much for excellent suggestion. As we discussed on the limitation of our study in the 6th paragraph in the discussion, and possible target of the application of myo-inositol in the 8th paragraph in the discussion. We are happy to be able to share our ideas and concerns with the reviewer. We are not sure whether we can add the separate subchapters or not, because our manuscript is just original article, but not review article. Therefore, we would like to follow the editor’s decision.

  1. “In the end of the discussion, you should also add the aspect of "Relationship between pain severity, satisfaction with life and the quality of sleep. (especially in patients with temporomandibular disorders), focusing on the multifactorial aspects of TMD and OSA - this is not only bite, occlusion, but also many factors and this should be at least one paragraph in the discussion.”

---Response:      We are very impressed by your deep insights. As the reviewer suggested, Skeletal Class II is not just the problem of intermaxillary relationships between maxilla and mandible, but it does induce several pathological symptoms including sleep and temporomandibular disorders. We added the further deeper discussion from the point of quality of life in the discussion.

  1. “I rate the paper high, although some corrections should be done Thank you.”

---Response:      Thank you very much for fruitful discussion and suggestion. We wish the revised manuscript meets the standards expected for publication.

Round 2

Reviewer 1 Report

Comments and Suggestions for Authors

Dear authors,

Thank you for revising the article.

My comments have been adequately addressed.

Comments on the Quality of English Language

The quality of the English is good, but minor linguistic changes are necessary.

Reviewer 2 Report

Comments and Suggestions for Authors

the authors have addressed all concerns well. I have no more concerns

Reviewer 3 Report

Comments and Suggestions for Authors

Dear Authors, although the corrections are not exactly what I ment, they are very smart and the improvemet of the paper is amazing. Thank you